# Challenges and Considerations in Diagnosing and Managing p16+-Related Oropharyngeal Squamous Cell Carcinoma (OPSCC) with Neck Metastasis: Implications of p16 Positivity, Tobacco Exposure, and De-Escalation Strategies

**DOI:** 10.3390/jcm13226773

**Published:** 2024-11-11

**Authors:** Giovanni Motta, Benedetta Brandolini, Tonia Di Meglio, Salvatore Allosso, Massimo Mesolella, Filippo Ricciardiello, Marco Bocchetti, Domenico Testa, Gaetano Motta

**Affiliations:** 1ENT Unit, Department of Mental, Physical Health and Preventive Medicine, University of Campania “Luigi Vanvitelli”, 80131 Naples, Italy; benedettabrandolini96@gmail.com (B.B.); tonia.dimeglio@studenti.unicampania.it (T.D.M.); domenico.testa@unicampania.it (D.T.); gae.motta@libero.it (G.M.); 2Otorhinolaryngology-Head and Neck Surgery Unit, Department of Neuroscience, Reproductive and Odontostomatological Sciences, University of Naples Federico II, 80138 Naples, Italy; salvatore.allosso@unina.it (S.A.); massimo.mesolella@unina.it (M.M.); 3Ear, Nose and Throat Unit, AORN “Antonio Cardarelli”, 80131 Naples, Italy; filipporicciardiello@virgilio.it; 4Department of Life and Health Sciences, Link Campus University, Via del Casale di San Pio V 44, 00165 Rome, Italy; m.bocchetti@unilink.it

**Keywords:** NCUP, OPSCC, p16 IHC, HPV testing, p16-positive HPV-negative OPSCC, p16-positive-HPV-positive OPSCC, tonsillectomy, tobacco exposure, p53, cystic neck metastasis, extranodal extension (ENE), radiotherapy, de-escalation, neck dissection

## Abstract

Patients presenting with cystic metastasis in the neck lymph nodes and no obvious primary tumor, neck cancer of unknown primary (NCUP), represent a very complex management challenge, especially today in the Human Papillomavirus (HPV) era. Given the increasing incidence of HPV-related oropharyngeal squamous cell carcinoma (OPSCC), further detection methods other than p16 IHC (immunohistochemistry) for HPV testing are crucial. An HPV-positive status can localize the tumor to the oropharynx, a common site for occult primaries. Furthermore, up to 15% of p16 protein-positive tumors are actually HPV-negative. Failure to perform additional HPV testing can have dangerous prognostic and therapeutic implications such as de-escalation strategies that hesitate in an undertreatment. The other important topic faced in this study is the role of smoking and p53 mutations, especially their significance in HPV-positive cancers and the role of extranodal extension (ENE) in HPV-positive patients. In this paper, biomolecular, diagnostic, prognostic and therapeutical aspects are critically analyzed to make a precise diagnosis and accurately estimate the prognosis of such patients.

## 1. Introduction

Patients who present with cystic metastasis in the neck lymph nodes and no obvious primary tumor have what is known as neck cancer from unknown primaries (NCUPs). This represents a complex diagnostic and management challenge, especially today in the HPV era. It is estimated that a significant number of these cases are related to p16-positive OPSCC (oropharyngeal squamous cell carcinoma), with the primary tumor often involving the palatine or lingual tonsils or the base of the tongue (BOT) [1,2,3,4,5,6,7,8]. Early neck metastasis could be explained by the fact that OPSCC, especially when HPV-related, often originates from the reticulated epithelium lining the tonsillar crypts, where the basal cell layer is incomplete and the supportive basal membrane is disrupted and non-contiguous. This lack of structural integrity can easily explain the early neck metastasis seen at presentation even when the primary tumor is not evident during clinical evaluation [9]. Neck metastasis is usually homolateral to the site of the primary cancer; however, it is possible for there to be the occurrence of neck metastasis on the contralateral side or, even more rarely, for synchronous bilateral SCC of the tonsils [5,8,10,11,12,13,14,15,16,17,18,19,20]. Imaging exams such as PET-CT are not always able to identify the primary oropharyngeal cancer, and they struggle to differentiate the small tumors from chronic inflammation [5,15,21]. Although NCUP is more common today in the HPV era, it remains a rare diagnosis, making it difficult to conduct large-scale trials. Currently, there is no internationally agreed-upon approach to managing these cancers [22,23,24]. Given the increasing incidence of HPV-related OPSCC, HPV testing is crucial as it can localize the tumor in the oropharynx, significantly impacting patient treatment [25,26,27,28,29]. However, even in cases where the tumor is HPV-related and the primary site is identified in the oropharynx, clinical, biomolecular and morphological factors must be carefully considered and are currently under study in order to accurately determine prognosis and the management of these subsets of patients [28,30,31,32,33,34,35,36,37,38,39,40,41,42,43,44,45,46,47,48].

This paper aims to critically analyze the most controversial diagnostic, prognostic and therapeutical aspects in p16-positive OPSCC with neck metastasis. Our research focuses on highlighting these critical aspects in two patient subtypes: p16 IHC (immunohistochemistry)-positive patients who are HPV-negative and p16 IHC-positive HPV-positive patients who are long-time smokers and/or drinkers and/or that may harbor the p53 mutation.

## 2. Materials and Methods

This review was conducted to investigate the biases in the diagnostic and therapeutic process of two OPSCC patient groups: p16-positive, HPV-negative patients, and p16-positive, HPV-positive patients with exposure to traditional risk factors and/or p53 mutations. The search strategy included a combination of Medical Subject headings (MeSH) terms and free-text words related to p16, HPV, oropharyngeal squamous cell carcinoma (OPSCC) and treatment biases. The keywords used for the literature research were as follows: NCUP, OPSCC, p16IHC, HPV testing, p16 positive HPV negative OPSCC, p16 positive HPV positive OPSCC, tonsillectomy, tobacco exposure, p53 mutations, cystic neck metastasis, extranodal extension (ENE), radiotherapy, de-escalation, neck dissection. The comprehensive literature review was conducted using the following electronic databases: PubMed, Scopus and Cochrane Library. Studies were included if they were original research articles reporting on patients with OPSCC. Studies were excluded if they were reviews, while case reports and original articles were excluded if insufficient data on the diagnostic and therapeutic pathways of the two patient subsets were described. The quality of the included studies was assessed using the Newcastle–Ottawa Scale for observational studies.

## 3. Results

HPV-positive OPSCC patients show specific clinical and radiological features. It is recommended to perform bilateral tonsillectomy (BT), rather than tonsillar biopsies, to identify the primary tumor. Detecting small tumors through a BT is crucial for determining the extent of the primary tumor (T extent), which in turn affects the subsequent treatments and facilitates easier follow-up due to the symmetry of the palatal arches. However, P16 IHC alone is not sufficient for diagnosing HPV-related OPSCC, and additional detection methods are necessary. The most effective method for detecting HPV is still a topic of debate. The influence of tobacco exposure and p53 mutations should be considered and further investigated in future studies, especially in cases of HPV-positive OPSCC. ENE must be taken into consideration in the prognostic staging of HPV-positive tumors as several studies have demonstrated its impact on survival. For N3 cases, primary surgical treatment involving neck dissection (ND) and BT followed by adjuvant radiation might be the optimal choice, considering the risk of irradiating p16+ tumors that might not respond to radiation (such as p16+HPV−, p16+HPV+ but smokers and p53 mutations). Diagnosis, prognosis and therapeutical implications must be addressed considering clinical, biomolecular and morphological aspects. To date, the numerous biases that still exist in the diagnostic and prognostic process make it challenging to consider de-escalation protocols.

## 4. Discussion

### 4.1. Identifying the Primary Tumor

What to do, where to look and why is it so important? P16+ OPSCC frequently presents with a cervical neck metastasis and the primary tumor is often not detectable at the first clinical and fiberoptic evaluation [29,49,50]. This is because p16-positive and presumptively HPV-related OPSCCs are often small cancers that emerge deep within the tonsillar crypts [51,52]. Specifically, it has been observed that the tumor develops in the reticulated epithelium lining the tonsillar crypts, where the basal cell layer is incomplete, and its supportive basal membrane is disrupted and not continuous. The lack of structural integrity can easily explain the early neck metastasis even when the primary tumor is not evident during clinical intraoral evaluation and upper airway fiberoptic examination [9,29,51]. Neck metastasis is usually homolateral to the primary site. However, contralateral metastasis, and the even rarer occurrence of synchronous bilateral tonsillar cancers, are possible and have been described. In 2010, Roeser et al. [13] reported a case of bilateral simultaneous metastatic palatine tonsil squamous cell carcinoma in a 51-year-old man. This patient presented with a cystic metastasis of the neck. In a 2009 retrospective study [14], Waltonen et al. evaluated the effectiveness of various diagnostic modalities in 183 patients with metastatic carcinoma of the head and neck from an unknown primary. The authors concluded that diagnostic procedures, including PET-CT and pan-endoscopy with directed biopsies (including BT), offer the best chance of finding the location of the hidden primary tumor. The neck levels mainly involved in neck metastasis were level 2 (78.2%) and level 3 (35.6%). In a retrospective investigation by Issing et al. published in 2003 [8] showing and analyzing the medical records of 167 patients admitted and treated for cervical NCUP, the authors recommended BT not only for diagnostic purposes. In a retrospective study by Sokoya et al. published in 2018 involving 190 patients [5], the authors also claim that BT should be performed. In this study, 17% of the detected tonsillar primary cases—3 cases out of 18—were either contralateral or bilateral to the presenting neck metastasis. In a case series published by Koch et al. in 2001 [10], out of 16 patients with neck metastasis, 2 patients had cancer in the contralateral tonsil, while another 2 had bilateral tonsillar primary cancers, reporting rates of contralateral spreads of metastatic cancer to be close to 10%. The neck levels mainly involved in the metastasis were the second and third levels of the neck and the metastasis had cystic appearance on imaging exams [1,2,3,4,5,6,7,8,9,10,11,12,13,14,15,16,20,53]. The cystic appearance and the frequent involvement of levels 2–3 in HPV-positive OPSCC are usually reported by various authors and may be characteristic features to consider during the diagnostic workup of NCUP. It is important to note that HPV-related OPSCC develops from the tonsil depths and subtle tonsillar asymmetry can be more noticeable to the radiologist than to the clinician during the initial evaluation. Therefore, these features should be taken into consideration as part of the initial diagnostic workup in order to guide the clinician [3,4,7]. Further diagnostic imaging exams such as PET-CT are not always able to identify the oropharyngeal primary sites and lack in distinguishing chronic inflammation uptake from cancer uptake. In 2018, Sokoya et al. [5] found that among 190 patients with head and neck squamous cell carcinoma, 87 were positive and 103 were negative on PET/CT, concluding that there is a high rate of cancer diagnosis even in PET/CT-negative patients and therefore site-directed biopsies of common sites should be performed and in particular BT. Another retrospective study by Liu et al. published in 2019 [54] involving 40 patients showed that FDG PET/CT detected the primary tumor in 40% of patients. In 2016, Mani et al. [55] revealed that among 52 patients with head and neck NCUP, twenty-seven PET/CT scans suggested a primary site. Three false-positive PET/CT scans were noted after pan-endoscopy, while another three confirmed OPSCCs were discovered on pan-endoscopy, which were undetected on PET/CT. Pencharz et al. [56] conducted a retrospective study in 2019 on 25 patients affected by tonsillar carcinoma. They noticed that a Standardized Uptake Value (SUV) ratio between tonsils > 1.6 was highly suggestive of squamous cell carcinoma. However, this value should only be used to determine the site of biopsy, as some malignant tonsils had normal FDG uptake [5,16,55,56]. This evidence suggests that imaging exams, and in particular PET/CT, performed before pan-endoscopy and biopsy, are not sufficient alone in detecting the primary site in the diagnostic workup of NCUP, as they can result in both false-positives and false-negatives. Nevertheless, they may be helpful in directing targeted biopsies of common sites of primary squamous cell carcinoma [54,56]. In the case of small tumors that usually invaginate in the tonsillar crypts and that are p16-positive and presumptively HPV-related, a diagnostic tonsillectomy ensures the radical removal of the tumor. This is probably the reason why, in the previously cited retrospective study by Issing et al. published in 2003 [8], the authors detected significantly improved survival rates in patients with NCUP where the primary cancer was then identified through a bilateral tonsillectomy in addition to ND and post-operative radiotherapy [29]. Unfortunately, today, the rates of primary detection are suboptimal, reported as being as low as 50% of detection. Because treatment and patient outcomes are largely dependent on the primary site identification, BT should be performed (Table 1) not only because primary site identification influences further eventual treatments, but also allows an easier follow-up and often means radicality on the T extent.

### 4.2. Morphologic Features of HPV-Related and HPV-Non-Related OPSCC

Another factor to add in the diagnostic and prognostic equation? Non-keratinizing squamous cell carcinomas (NKSCCs) are less complex at a biomolecular level and have a better prognosis because they are HPV-positive even though they may appear less differentiated. On the other hand, keratinizing squamous cell carcinomas (KSCCs), even if they look more differentiated, typically show worse outcomes and are linked to traditional risk factors in HPV-negative patients. Even hybrid variants have been described in various papers [9,26,57,58,59,60]. However, it is not always simple to differentiate between them, and the identification could represent a real challenge, especially when dealing with the significance of hybrid variants. In a combined retrospective/prospective data collection published by Chernock et al. in 2009 [30], 118 OPSCCs were included. The authors showed that overall survival and disease-specific survival rates varied among histologic groups. NKSCC had better overall survival and disease-free survival compared to KSCC due to the strong association between NK SCC variants with HPV and K SCC variants with traditional risk factors. On the other hand, hybrid SCC had an intermediate survival rate. While some patients with hybrid variants may have good outcomes, others may not. It is plausible that HPV-negative hybrid variants may have worse outcomes compared to NK HPV-positive SCC because the keratinizing part of the hybrid variant might be the expression of tobacco exposure and/or p53 mutation, but more specific research is needed to address this issue [30,43,61]. Similar histologic morphologies were also described by Fujimaki et al. in a retrospective paper published in 2013, which included 66 cases [60]: the findings revealed that NKSCC and hybrid SCC were more likely to be HPV-associated, while KSCC tended to be HPV-un-associated. However, this study did not focus on the relationship between patients’ outcomes and morphology and there was no information on the relationship between hybrid variants and smoker patients [60]. Focally keratinizing areas in the subset of basaloid and non-keratinizing morphologies were also reported by other authors [62]. Further studies on hybrid tumors are required in order to identify the factors responsible for bad outcomes. On the other hand, when a tumor is clearly keratinizing or non-keratinizing, morphology might be considered for diagnostic purposes and to predict patient outcomes. In a retrospective study published in 2008 [26], Zhang and colleagues describe the specimens with NKSCC morphology and HPV positivity as metastatic tumors from the oropharynx. The previously cited work by Chernock et al. in 2009 stressed that NKSCC had better overall survival and disease-specific survival when compared to KSCC. In a retrospective study by Molony et al. published in 2020, which included 168 patients, the role of tumor morphology was tested in assigning HPV status in OPSCC [30]. Morphology was classified as KSCC and NKSCC. P16 was performed and additional HPV DNA PCR was performed in cases where morphology and p16 were discordant. The authors found that a combination of morphology and p16 IHC, with additional HPV testing for discordant cases, was the most accurate predictor of overall survival and disease-specific survival in OPSCC. Therefore, the authors suggest that using morphology to select cases for additional HPV testing may be practical and cost-effective [26,30,63]. Having presented such evidence, what might be true is that HPV-unrelated OPSCCs have a bigger T extent, while tumors with a big N extension at presentation with a micro-T are then identified in the tonsillar crypts or BOT are more commonly HPV-related NK OPSCC [26]. Therefore, if a hybrid variant is found in the neck nodes, the initial evolution might have been HPV-driven but it would be very important to investigate the relationship between the keratinizing share of the tumor, traditional risk factors exposure (tobacco and/or alcohol) and/or p53 mutation in order to predict hybrid tumors with the worst prognosis. Further studies must clarify these issues, linking the clinic–prognostic characteristics of the tumor to histology and to molecular biology. The factors to study and consider in the presence of p16-positive neck metastasis, including morphology, are listed in Table 2. Requiring the knowledge and skills of various figures, a multidisciplinary approach is essential in these cancers.

### 4.3. Diagnostic and Prognostic Aspects Related to P16 and HPV Discordance

Are further detection methods required? It is known that HPV-related OPSCCs have a better radiotherapy response, better survival rate and overall better prognosis when compared to HPV-negative OPSCCs [57,58,59]. For this reason, the eighth edition of the American Joint Committee on Cancer and the Union for International Cancer Control has now separated the staging systems for these two malignancies, with very important prognostic–therapeutical implications, such as de-escalation protocols to reduce the toxicity of standard multimodal treatments [59,60,61]. However, this is not always based on the presence of the HPV, but on the IHC positivity to the p16 protein, a surrogate of the HPV transformative infection [2,62,63,64,65]. Although p16 is a good surrogate of high-risk HPV infection, it is not a direct demonstration of the presence of the virus, since it can be overexpressed by other factors, with a significant number of false-positives. In fact, it is estimated that up to 15–20% of p16-positive OPSCCs are HPV-negative, with the risk that p16-positive HPV-negative patients could represent a distinct subgroup with different and worse outcomes. In a 2013 retrospective study by Stephen et al., the authors investigated the role of p16 in 10 HPV-positive and 10 HPV-negative OPSCCs and emphasized that the p16-positive HPV-negative cases were likely a distinct subgroup that showed a substantial deviation from the p16-positive HPV-positive referent control group, suggesting a distinct subgroup lacking any HPV genotype. In a 2018 retrospective study by Nauta et al. [66], 1204 OPSCC patients were included, and it was seen that out of the 388 p16-positive patients, 48 patients (12.4%) were HPV-DNA-negative. More importantly, this subgroup revealed a significantly worse five-year survival compared with the HPV-DNA-positive cancers. The authors concluded by encouraging the importance of additional HPV testing. Similar findings and conclusions, based on 238 OPSCC cases, were supported by Sathasivam et al. in 2018 [67]. In a 2020 retrospective study by Wagner et al. [68], which investigated p16 expression as a single marker to select patients for de-escalation, the authors found that 16.6% of p16-positive OPSCCs were actually HPV-DNA-negative, concluding that p16 IHC alone is insufficient to identify patients who can benefit from treatment de-escalation. In a recent retrospective study published in 2023 by Lu et al. [69], the authors noticed that among 190 HPV-positive OPSCCs, p16 IHC misclassified 14% of the samples and therefore the authors recommend testing OPSCC patients for HPV but not p16. A very intriguing retrospective study was published in 2021 by Benzerdjeb et al. [62] and the authors combined p16 and p53 IHC to predict HPV tumor status in 110 patients. The main findings were that the p16-positive/wild-type p53 cases were significantly associated with HPV-positive tumor status (50 of 110). Interestingly, 13.5% of p16-positive cases were unrelated to HPV and showed p53 mutant type [39,45,58,59,60,61,62,63,64,65,66,67,68,69,70,71,72,73,74,75]. This is why several authors stress the importance of further detection methods such as adding in situ hybridization (ISH), HPV DNA PCR, assessment of E6/E7 mRNA and HR-HPV RNA scope or even leaving HPV confirmatory testing in patients where p16 status and morphology are not coherent. In a case–control study by Pannone et al. published in 2012 which included 22 OPSCCs, the authors used a combined triple method, which included IHC, ISH and PCR techniques to detect causative HPV. Out of the 22 OPSCCs, 21 were HPV-positive by IHC. Out of these 21, 7 were also positive for HPV-DNA and 6 were also positive via the PCR technique. Such results indicate that p16 IHC is not sufficient when used as the only HPV-detecting method. In a 2012 retrospective study by Bishop et al. [76,77] that included 77 oropharyngeal cancers, 25 cases had discordant results of being p16-positive and HPV DNA-negative. In this discordant group, the presence of E6/E7 transcripts was detected in 21 out of 25 patients (84%). Following these results, the authors suggest that testing for HPV E6/E7 transcripts by RNA ISH is ideal because it confirms the presence of integrated and transcriptionally active viral particles, allowing the visualization of viral transcripts in tissues, thus being technically feasible for routine testing. In a retrospective paper by Rietbergen et al. published in 2013 [78], the authors evaluated a test algorithm on a series of 240 OPSCC samples. The algorithm, which involved a combination of p16 IHC followed by HR HPV DNA PCR in positive cases, showed an accuracy of 98% and it was tested against the gold standard represented by the detection of viral mRNA carried out by quantitative RT-PCR. Another retrospective study published by Schache et al. in 2013 [79] and including 79 OPSCCs compared HR- HPV RNA scope with the reference test represented by qRT PCR for HR-HPV. The results showed excellent analytical and prognostic performance against the reference test. Similar results were identified on 50 OPSCCs by Mirghani et al. in 2015 [80]: the authors tested 50 OPSCCs using the RNA scope HPV test, p16 IHC and chromogenic ISH for HR HPV DNA. The results were compared to those of quantitative RT-PCR and showed that the RNA scope HPV test demonstrated excellent analytical performance against the gold standard, and it is easier to interpret compared to chromogenic ISH. In a case–control study by Rooper et al. published in 2016 [81], which included 82 head and neck squamous cell carcinomas, the authors evaluated whether E6/E7 mRNA ISH (RISH) could detect transcriptionally active HPV in 42 discordant cases that were p16-positive but HPV DNA-negative by ISH. The results were that RISH identified E6/E7 mRNA in 37 (88%) p16-positive DISH-negative cases and concluded that RISH should be considered as a first-line platform for the determination of HPV status in OPSCCs. In a retrospective comparative study published by Bauwens et al. in 2021 and involving 263 patients, the prevalence and distribution of neck nodes metastasis in HPV-positive and in HPV-negative patients were compared and the HPV status was assessed by p16 IHC. In this study, only in cases of discordance between p16 and morphology was ISH with a high-risk HPV DNA probe performed to address the HPV status. In the previous paper published by Lu et al. in 2023 [69], the authors suggest assessing HPV status by HR-HPV RNA ISH with this technique being relatively simple in time and costs and requiring minimal tissue quantity when compared to the PCR approach. Important prognostic data come from a retrospective study published by Shinn et al. in 2021 [82] and including 70 OPSCCs: in this paper, the authors reported that, in univariate survival analysis, patients with discordant p16 and HPV mRNA had significantly greater rates of death and disease recurrence/treatment failure compared to those with concordant positive results [2,65,67,68,75,76,78,83,84,85]. Considering all these data and given the numerous diagnostic and prognostic implications, further detection methods for HPV identification must be performed. Even though many authors are very confident in considering RISH the gold standard for detecting the presence of clinically relevant HPV (Table 3), it is possible to have discordant results even between various additional detection methods implied in clinical practice and therefore investigating which is the best detection method remains a discussed topic.

### 4.4. The Role of Tobacco Exposure and p53 Mutations in HPV + OPSCC

Prognostic and therapeutic aspects: Recent evidence has shown that, nowadays, about 31% of HPV OPSCCs occur in heavy smokers and drinkers and therefore exposure to tobacco may be reflected both at a morphological and biomolecular level [43,48,61,64,86]. Even when the tumor is actually HPV-related, carcinogenesis might be driven by both tobacco-related and HPV-related pathways. In 2000, Gillison et al. published a paper in which 259 patients with head and neck squamous carcinomas were included. In this study, some patients with HPV-positive status and p53 mutations were present, but the exact number of patients is not specified. In a study published in 2007 in the New England Journal of Medicine, which included 100 OPSCC patients, of which 56 were smokers and HPV-positive, D’Souza et al. showed through multivariate logistic regression models that tobacco use was associated with an increased risk of developing HPV-positive OPSCC. Similarly, in a case–control study by Smith et al. published in 2010 that included 237 cancers, the authors studied the individual and joint effects of tumor markers for differences in predicting head and neck cancer survival. Overexpression of p16 and p53 occurred in 38% and 48% of head and neck tumors. High-risk HPV was detected in 28% of tumors. The worse prognosis was found in the tumors that were p53-positive or HPV-negative [45,57,62,86,87]. Hence, investigating the relationship between tobacco exposure, morphology and p53 mutations in HPV-positive OPSCC might be crucial for the correct establishment of the diagnostic, prognostic and therapeutic process. There are plenty of data demonstrating the effects of such traditional risk factors in inducing the mutation of the onco-suppressor protein p53 [45,46,62,87]. Usually, p53 mutations are present in patients with a smoking and drinking history and are much less common in patients who neither smoke nor drink, but can still be found, so it might be appropriate to screen for p53 mutations even in patients without exposure to the traditional risk factors like Benzerdjeb et al. did in their experience [62]. In a prospective analysis published by Brennan et al. in the New England Journal of Medicine in 1995, the association between cigarette smoking and mutation of p53 was studied in 129 patients. The frequency of p53 mutation differed among patients based on their smoking status and drinking habits: in fact, patients who both smoked and drank had an incidence of p53 mutation of 58%, patients who only smoked had an incidence of p53 mutation of 33% and patients who neither smoked nor drank had an incidence of p53 mutation of 17% [46]. Even if rare because they are usually non-overlapping events, HPV positivity and p53 mutations are a clinical situation that can coexist. In a case–control study published by Smeets et al. in 2006, which included 146 head and neck cancers including OPSCC, the authors observed that out of the 12 cases in the HPV-positive group, 10 cases showed p53 mutations. In a multicenter retrospective study published by Westra et al. in 2008, out of 12 HPV-positive OPSCC, 3 (25%) even had p53 mutations [9,45,47,57,62,88]. For such reasons, the role of traditional risk factors and p53 mutations often strictly related to those are fundamental aspects to discuss and highlight in the prognostic stratification of HPV-positive patients. The p53 protein is present in the down-regulated not mutated wild-type form in the classical young non-smoker HPV-positive patient. Therefore, it might be possible that the young age, together with the absence of risk factors and p53 mutations, can explain why these patients are more radiosensitive and have a better prognosis. These facts are demonstrated by Lindel et al. In a retrospective study published in 2001 that included 139 patients. A total of 14 out of 139 (19.46%) cases were HPV-positive OPSCC with no mutated p53. The OPSCC HPV-positive patients not p53-mutated had better outcomes in terms of local control, disease-free survival and overall survival [89,90,91]. It is becoming clearer that genetic mutations that are usually tobacco-induced, and in particular the p53 mutation, can negatively influence the prognosis of HPV-positive OPSCC and make, in general, head and neck squamous cell carcinomas less responsive to therapy, compromising the outcomes of such patients. In a retrospective study published by Ma et al. in 1998 that included 50 patients (15 OPSCCs), the authors observed that 45% of tumor samples had p53 mutations, which were associated with loco-regional recurrence. The risk of loco-regional recurrence was significantly higher in patients with p53 mutations compared to those with wild-type p53. The study also found that p53 gene mutations can differ in recurrent tumors compared to the original primary tumor, suggesting their potential role in the differential diagnosis of a second primary. In a prospective multicenter study published in the New England Journal of Medicine by Poeta et al. in 2007 that included 420 patients with SCC (224 with p53 mutations), it was shown that p53 mutations were associated with decreased overall survival. The study also identified other significant prognostic factors for survival such as tumor stage, nodal stage, smoking history and alcohol abuse. In a prospective clinical trial published by Fakhry et al. in 2008, which included 38 HPV-positive OPSCCs, the authors documented that HPV-positive cancers had better survival than HPV-negative cancers. However, not all 38 HPV-positive OPSCCs had good outcomes; there were a total of seven deaths, but the authors did not specify if these patients were smokers or harbored p53 mutations. In the previously cited paper by Gillison et al. published in 2000, patients with p53 mutations had different survival outcomes compared to those without p53 mutations. In a retrospective study published in 2019 by Ziai et al., and in which 160 cases were included, the authors determined that Bcl-xL and p53 together trended towards patterns of outcome and survival, dependent on HPV status and smoking status. This trend was statistically significant in patients who were non-smokers or those with p16-positive disease. Similar patterns of survival with improved disease-specific survival outcomes in patients with low p53/Bcl-xL were demonstrated by Kumar et al. in 2007 [40,41,42,43,44,45,57,78,89]. Moreover, it is fundamental to mention another retrospective study by Koch et al. published in 1996, which included 110 patients. The authors documented the fact that head and neck squamous cell carcinomas, which harbored p53 mutations (48 out of 110 patients), and that were treated with radiation therapy, had higher rates of loco-regional failure, once again strengthening the hypothesis that p53 mutations render a squamous carcinoma less responsive to radiotherapy. A consideration of such cases can be that if a tumor is surgically resectable and has p53 mutations, radiation therapy might not be the most appropriate strategy as a primary treatment, even because options after radiation treatment failure are very limited and poorly beneficial in terms of long-term outcomes for the patients [89,92]. As a matter of fact, in a randomized controlled trial published in 2010 in the New England Journal of Medicine, which included 743 patients with OPSCC, the authors were able to stratify the patients into three risk groups: interestingly, and as now expected, HPV OPSCC patients who were heavy smokers or had high nodal stage did not fall into the low-risk group, despite their HPV positivity [39]. The p53 status of the patients was not investigated in this randomized controlled trial (RCT) but it might be possible that the HPV patients that were not in the low-risk group could have harbored such mutations because tobacco smoking significantly increased the risk of death in OPSCC patients and this effect was similar for both HPV-positive and HPV-negative patients. The evidence presented by these studies demonstrated that tobacco exposure and/or p53 mutations negatively influence the prognosis of head and neck SCCs. However, few authors have specifically investigated this issue specifically relating it to HPV-positive OPSCC [39,45] (Table 4). Hence, it is essential to identify—among HPV-related OPSCC—certain factors that could negatively influence the outcomes of these patients.

### 4.5. The Prognostic Role of ENE+ in p16-Positive (HPV-Negative and HPV-Positive) OPSCC

Why should it be considered? For p16-positive OPSCCs, another important controversial aspect is the fact that ENE+ is not considered in the pathological staging system of p16-positive OPSCC. To show that the diagnosis of HPV cases must be more accurate and secure, it is fundamental to highlight that the prognostic irrelevance of ENE has been evaluated only by retrospective studies, with follow-ups that could have been longer [93]. The prognostic significance of ENE+ as a possible risk factor even in HPV-positive OPSCC remains a discussed topic in many current studies [36,37]. Especially in the last few years, several papers have emphasized the fact that p16-positive and HPV-related tumors with ENE+ have poorer outcomes (Table 5). Already in a retrospective study published in 2011 by Lewis et al., which included a p16-positive cohort (101 cases included and 90% were p16-positive), the authors brought to attention that the 37 patients with grade 4 ENE—defined as having no residual nodal tissue or architecture—had poorer outcomes. In a retrospective study by Spector et al. published in 2014, in which 156 patients with HPV-positive OPSCC were included, the authors reviewed the CT scans of patients who underwent chemoradiation and found that the patterns of nodal metastasis were significantly associated with overall survival and disease-specific survival. Furthermore, the researchers propose a new staging system that considers the size, bilaterality and matted nodes—defined as three nodes abutting one another with the loss of an intervening fat plane that is replaced with evidence of extracapsular extension—which argues more accurately the survival differences. This new classification system showed improved risk stratification for the nodal classification system. In 2015 and 2016, both Huang et al. and O’Sullivan et al. proposed staging systems for the improvement in survival predictions [94,95]. However, both studies did not discuss specifically the impact of ENE+ in HPV-positive OPSCC. This issue was specifically addressed by An et al. in a retrospective study published in 2017, which included 1043 HPV-positive OPSCCs and nearly half of these had ENE+ (43.5%) [93]. The study testified that ENE+ was associated with worse overall survival in HPV-positive OPSCC. Similarly, in a retrospective study by Beltz et al. published in 2019, which included 255 OPSCCs (105 HPV-positive), the authors reported that HPV-related tumors that were also ENE+ had significantly worse overall survival. Another retrospective study by Freitag et al. published in 2020, which included 92 patients (62 ENE+), attested that ENE and HPV 16 DNA status were independent predictors of impaired survival in p16-positive OPSCC, concluding that these two factors must be included in the prognostic staging of p16-positive OPSCC. Consistent with these last studies, in a multicenter retrospective study published in 2020 by Bauer et al., in which 3138 HPV-positive OPSCCs were included, the authors showed that ENE-negative cases had the highest 5-year survival, while ENE-positive cases had the lowest 5-year survival and almost twice the hazard of death compared to ENE-negative cases, concluding with the recommendation to consider ENE in staging and treatment decision making for HPV-positive OPSCCs [32,36,37,38,92,93,96]. In this context, the most appropriate strategy to treat N-positive patients with and without ENE might be surgical through a unilateral neck dissection followed by adjuvant therapy. In the previously cited paper by Beltz et al. published in 2019, the authors also found that for the subgroup of patients with HPV-positive OPSCC, primary surgery was one of the factors leading to significantly better overall survival [37]. Two papers published in 2017 by Zenga et al. have shown that a primary surgical approach to N3 cases guaranteed favorable long-term survival and regional control [34,35]. More importantly, as Wichmann et al. demonstrated in their paper, a surgical approach to the neck through a ND and consequent histological examination of the surgical specimen prevents underdiagnosis of ENE, allowing the most adequate post-operative treatment to the patients [35]. Based on these data, it is important to stress the fact that, before discussing the prognostic role of ENE and HPV status, a more precise diagnosis of HPV cases is mandatory. When the patient is surgically treatable, a primary surgical approach is advisable with transoral resection and ND, followed by adjuvant therapy in N3 cases. This strategy could be optimal for both diagnostic and therapeutic reasons. It might be a risk to use primary radiation therapy in a tumor that, at a biomolecular level, is not responding adequately to radiation, especially because the therapeutical options after a radiotherapy failure are very limited and do not guarantee long-term outcomes.

### 4.6. De-Escalation Protocols

Are we ready today? For the same reasons, today, without a more precise diagnosis, it is inappropriate and risky to propose de-escalation protocols for HPV-positive OPSCC, as demonstrated by important clinical trials. In the randomized controlled trial published by Gillison et al. in 2019, radiotherapy plus cisplatin (406 patients) was superior in terms of overall survival and cancer control compared to radiotherapy plus cetuximab (399 patients). Similarly, in the one published by Mehanna et al. in 2019, cisplatin-based chemo-radiotherapy (166 patients) had better overall survival, lower rates of recurrence and metastasis and better tumor control compared to the cetuximab-based bio-radiotherapy [97,98]. Furthermore, it was proven that cisplatin chemoradiotherapy offered a better quality of life and it was less expensive when compared with cetuximab bio-radiotherapy [99]. Other important trials to mention are the AVOID trial, a phase 2 clinical trial, which de-intensified adjuvant therapy by avoiding primary site radiotherapy in patients with HPV-positive OPSCC who had safe margin resection with transoral robotic surgery (TORS) and ipsilateral neck dissection. The authors concluded that sparing the primary site and targeting only the at-risk neck may be safe but further studies are required to confirm this strategy [100]. The Orators trials compared de-intensified definitive radiotherapy to TORS + ND with de-intensified adjuvant radiotherapy and omission of chemotherapy. The results of these trials have not yet redefined the standard of care for HPV-associated OPSCC [101]. Interestingly, a phase 2 clinical trial of de-intensified chemoradiotherapy for patients with HPV OPSCC by Chera et al. showed favorable clinical outcomes but highlighted the need for further research in order to optimize patient selection and reduce toxicity [102]. De-escalation protocols are currently not recommended in clinical routine, less than ever in patients with only p16 IHC positivity [68,98,99,103]. In these conditions, without proper diagnosis implementation, the biggest risk is the one of compromising survival by undertreating oncologic patients [103,104]. Expectantly, in order to identify patients who may and may not benefit from de-intensification strategies, biomarker-driven risk stratification will be required to implement p16/HPV status and T-N categories [105].

## 5. Conclusions

Up to 15% of p16+ OPSCC cases are actually HPV-negative. The remaining are HPV-positive but may occur in patients with traditional risk factors, especially tobacco exposure. Therefore, a more accurate diagnosis with further HPV detection methods is required in order to adequately stage and manage p16+ OPSCC neck metastasis. The role of tobacco exposure and/or p53 mutations must be taken into consideration in p16+ OPSCC, and especially that a very careful evaluation and study of HPV-positive OPSCC occur. Until the diagnosis is more accurate, ENE+ should be considered even in p16+HPV+ OPSCC. Primary surgery with unilateral ND and BT might be the treatment of choice, given the numerous diagnostic and prognostic pitfalls, and for a better and easier follow-up. For all these reasons, today, it is inappropriate and risky to propose de-escalation protocols in clinical routine for the risk of undertreatment.

## Figures and Tables

**Table 1 jcm-13-06773-t001:** Authors who advise BT for primary tumor detection in the case of neck metastasis from unknown primary.

Author	Year	Journal	Study Design	Number of Patients Included in the Study
Roeser et al. [13]	2010	Laryngoscope	Case report and literature review	1
Waltonen et al. [14]	2019	Arch Otolaryngol Head Neck Surg	Retrospective study	183
Issing et al. [8]	2003	Eur Arch Otorhinolaryngol	Retrospective study	167
Sokoya et al. [5]	2018	Laryngoscope	Retrospective study	190
Koch et al. [10]	2001	Otolaryngol Head Neck Surg	Case series	16

**Table 2 jcm-13-06773-t002:** Factors to consider for decision making in case of p16+ OPSCC neck metastasis.

Clinical	Pathological
1. Exposure to traditional risk factors (tobacco, alcohol) or professional exposure	1. Further HPV detection methods
2. cENE	2. P53 mutational status
	3. Morphology
	4. pENE

**Table 3 jcm-13-06773-t003:** Authors advising further detection methods other than p16 IHC to identify HPV-related OPSCC.

Author	Year	Journal	Study Design	Numerosity	Further Detection Method Recommended
Pannone et al. [77]	2012	Infect Agent Cancer	Retrospective case–control study	22 OPSCC	Combined triple method: IHC, ISH and PCR
Bishop et al. [76]	2012	Am J Surg Pathol	Retrospective cohort study	77 OPSCC	RNA in situ hybridization (RISH)
Stephen et al. [75]	2013	Cancer Clin Oncol	Retrospective cohort study	80 SCC	Not provided by authors
Rietbergen et al. [78]	2013	Int J Cancer	Retrospective cohort study	240 OPSCC	p16 IHC followed by HR HPV DNA PCR
Schache et al. [79]	2013	Br J Cancer	Retrospective cohort study	79 OPSCC	RNA in situ hybridization (RISH)
Mirghani et al. [80]	2015	Mod Pathol	Retrospective cohort study	50 OPSCC	RNA in situ hybridization (RISH)
Rooper et al. [81]	2016	Oral Oncol	Retrospective case–control study	82 OPSCC	RNA in situ hybridization (RISH)
Nauta et al. [66]	2018	Ann Oncol	Retrospective cohort study	1204 OPSCC	Not provided by authors
Benzerdjeb et al. [62]	2021	Histopathology	Retrospective cohort study	110 OPSCC	Combined p16/p53 IHC
Wagner et al. [68]	2020	Br J Cancer	Retrospective cohort study	620 OPSCC	Not provided by authors
Lu et al. [69]	2023	Oral Dis	Retrospective cohort study	620 OPSCC	RNA in situ hybridization (RISH)

**Table 4 jcm-13-06773-t004:** Authors showing that tobacco exposure and/or p53 mutations negatively influence the prognosis of H&N SCC including HPV-positive OPSCC.

Authors	Year	Journal	Study Design	Numerosity
Koch et al. [91]	1996	J Natl Cancer Inst	Retrospective cohort study	110 SCC
Ma et al. [41]	1998	J Cancer Res Clin Oncol	Retrospective cohort study	50 SCC (15 OPSCC)
Poeta et al. [40]	2007	NEJM	RCT	420 SCC (93 OPSCC)
Ang et al. [39]	2010	NEJM	RCT	743 OPSCC
Smith et al. [45]	2010	Infect Agent Cancer	Retrospective cohort study	237 SCC

**Table 5 jcm-13-06773-t005:** Authors advising that ENE+ should be considered in HPV-positive OPSCC.

Author	Year	Journal	Study Design	Numerosity
Lewis et al. [96]	2011	Mod Pathol	Retrospective cohort study	101 OPSCC
An et al. [93]	2017	Cancer	Retrospective cohort study	1043 OPSCC
Beltz et al. [37]	2019	Eur J Surg Oncol	Retrospective cohort study	255 OPSCC
Freitag et al. [32]	2020	Cancer	Retrospective cohort study	92 OPSCC
Bauer et al. [36]	2020	The Laryngoscope	Retrospective cohort study	3138 OPSCC

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
