# Peer review of "Challenges and Considerations in Diagnosing and Managing p16+-Related Oropharyngeal Squamous Cell Carcinoma (OPSCC) with Neck Metastasis: Implications of p16 Positivity, Tobacco Exposure, and De-Escalation Strategies"

_jcm, 2024, doi:10.3390/jcm13226773_

Round 1

Reviewer 1 Report

Comments and Suggestions for Authors

The current manuscript describing challenges associated with making diagnoses and treatment strategies for patients with neck mets with NCUP is comprehensive and well written the authors present a clear narrative review of current and relevant literature and clearly delineate p16+/HPV- and p16+/HPV+. The review is well-written and will be of interest to a broad group of JCM readers. 

Author Response

Dear Reviewer,

We would like to express our sincere gratitude for your positive comments on our manuscript and for taking the time to carefully review it. We really hope that our research will make a valuable contribution to the field and benefit the broader scientific community. Thank you again for your time and expertise.

Giovanni Motta

Reviewer 2 Report

Comments and Suggestions for Authors

The paper is very well-written and very informative, and the English is very clear. HPV OPSCC de-escalation is still a topic of debate, and this paper does a good job of showcasing the different clinical scenarios where this is still an issue, especially in NCUP.

The Discussion is very well organized and clearly divided by each discussion topic. However, I'd suggest breaking some of these subsections into smaller paragraphs to digest the information more easily and better. Some of the sections were too long to be placed as one huge paragraph chunk. 

Extranodal extension under the Abstract (line 49) and the Methods section (line 112) is abbreviated as 'ECE.' Did you mean ENE as it was abbreviated in the rest of the manuscript? If I can also suggest defining 'RCT' in line 480 section 4.4, and 'SUV' in line 195 under section 4.1. 

Minor revision, HPV should be written as 'HP virus' or just 'HPV,' not 'HPV virus' as in line 289 in section 4.3, since the 'virus' portion would become redundant. I noticed the comma (,) was used to define decimal points (European way) instead of the period (.). And also very minor, but noticed that the page numbers are off in the manuscript. 

Author Response

Dear Reviewer,

Thank you very much for your positive comments on our work. As you correctly pointed out, we have addressed all minor revisions (ENE, RCT, SUV,HP virus, period instead of comma). However, after careful consideration, we have decided to maintain the current paragraph structure. We believe that this structure is a strength of our work. We have carefully considered the structure of these paragraphs, which resulted from a previous reorganization. While we explored the possibility of further subdividing some of the sections, such as the section on tobacco exposure and the p53 mutation, we ultimately decided that the current structure better represents the intricate relationship between these two factors. Given the frequent, yet not always concurrent, nature of these events, we believe that a combined discussion provides a more comprehensive overview. Thank you again for your valuable feedback and contributions to improving our work.

Reviewer 3 Report

Comments and Suggestions for Authors

Dear authors,

I rate the article submitted for review highly. Both the research problem, which is becoming increasingly serious at the moment, and the way the review is presented are very important medically and scientifically interesting.

Author Response

Dear Reviewer,

We would like to express our sincere gratitude for your positive comments on our manuscript and for taking the time to carefully review it. We really hope that our research will make a valuable contribution to the field and benefit the broader scientific community, particularly those involved in the treatment of these complex tumors. Thank you again for your time and expertise.
